# Low-Cost Sensors for Determining the Variation in Interior Moisture Content in Gypsum Composite Materials

**DOI:** 10.3390/ma13245831

**Published:** 2020-12-21

**Authors:** Daniel Ferrández, Carlos Morón, Pablo Saiz, Evangelina Atanes-Sánchez, Engerst Yedra

**Affiliations:** 1Departamento de Ingeniería de Organización, Administración de Empresas y Estadística, Universidad Politécnica de Madrid, 28660 Madrid, Spain; daniel.fvega@upm.es; 2Departamento de Tecnología de la Edificación, Universidad Politécnica de Madrid, 28040 Madrid, Spain; e.yedra@alumnos.upm.es; 3Departamento de Economía Financiera, Contabilidad e Idioma Moderno, Universidad Rey Juan Carlos, 28032 Madrid, Spain; pablo.saiz@urjc.es; 4Departamento de Ingeniería Mecánica, Química y Diseño Industrial, Universidad Politécnica de Madrid, 28012 Madrid, Spain; evangelina.atanes@upm.es

**Keywords:** low-cost sensors, Arduino, non-destructive evaluation, building materials, gypsum and plaster

## Abstract

Non-destructive testing can be used to determine some of the most relevant physical properties of building materials. In this work, two low-cost measuring devices were developed capable of determining the variation in real-time of the percentage of humidity that is produced in the construction of gypsum and plaster during the hardening process. For this, an Arduino resistive sensor and a capacitive sensor of our design were used. The results show how it is possible to determine the variations in mixing water content during the seven days of curing established by the UNE-EN 13279-2 standard as well relate to the mechanical resistance of the test specimens with the same percentage of humidity. Additionally, the study was completed with the determination of the formation of the dihydrate compound linked to this setting process in the test specimens by conducting X-ray diffraction and thermogravimetric analysis tests at different ages of the samples.

## 1. Introduction

Gypsum is one of the most widely used construction materials in the building sector, being the oldest artificial conglomerate known to mankind [1]. It is a material that has been present in all periods of history [2], from its traditional use as an interior cladding due to the fact of its qualities as a hygrothermal regulator [3] to its versatile current construction applications for the manufacture of interior partitions, decorative plaster moldings or lightened plates for ceilings among others [4]. However, its relative abundance, ease of working on site, and low cost compared to other conglomerates continue to favor its direct use today [5]. In addition, this traditional application generally requires increasingly scarce qualified labor [6]; for this reason, current research lines tend to reduce the limitations of gypsum as a construction material, which implies increasing its mechanical resistance, promoting industrialization through new construction systems, reducing the weight of precast to reduce building loads, and advancing the understanding of setting times [7,8].

This work revolved around the understanding of the hardening processes of gypsum composite materials and the study of the variation that occurs in the water content from mixing during this process. Several studies have focused on the determination of an adequate water/conglomerate ratio to obtain, in optimal conditions of workability of the material in a fresh state, the best performance in terms of achieving mechanical resistance [9,10]. There are studies, such as the one carried out by Ochoa et al. [11], in which by using natural additives extracted from *Agave* plants, the application times of the gypsum materials were increased without detriment to their mechanical properties, although it is true that most of the commercial additives to accelerate or delay the setting of this type of binding materials are usually composed of chemical compounds designed for this purpose [12,13]. However, this typology of studies on the hardening and setting of gypsum materials have generally focused more on the achievement of new dosages to improve certain properties than on monitoring and continuous monitoring of the setting process. This has led to the fact that most of the tests were carried out on days set by the reference regulations, evidencing the continuous variations in humidity experienced by the samples, their relationship with the variations in the physical properties of the material, and the performance in most studies of only some complementary chemical tests to study the evolution of the dihydrate and the hemihydrate [14,15].

In this sense, the measurement techniques used to determine the moisture content in construction materials have undergone notable progress in recent years [16]. In this way, it has evolved from the traditional gravimetric techniques based on the determination of moisture content through the weight variation that occurs in the material during its hardening and drying [17] to modern georadar techniques capable of recording the variations in the speed and amplitude of the waves when they propagate through a wet material [18]. Better known than microwave reflection [19] or nuclear magnetic resonance [20] is the use of hygrometers to determine the moisture content in porous soil and materials [21]. Among the commercial hygrometers, we found two well-differentiated types:(1)resistive ones that base their operation on the fact that, in contact to water, material increases its electrical conductivity [22](2)the capacities that obtain their response as a function of the variation in the dielectric properties of the material [23].


Regarding the second, their greatest advantage is that they do not physically penetrate the material, which makes them ideal for rehabilitation work [24]. In historical buildings where interventions are difficult, infrared thermography has been used to detect humidity with a high degree of precision [25].

Special mention requires the development of new measurement systems driven by Arduino technology for testing in the last decades [26,27]. In the field of moisture determination and its application in the building sector, we currently find a wide variety of studies. Some authors have used resistive sensors to better understand the setting processes of cement mortars made with recycled aggregates [28], although the use of this type of sensor to determine soil moisture and build automatic irrigation systems also stands out [29]. There are also studies using capacitive sensors in Arduino, such as the one carried out by Calva et al. [30], to determine the moisture content in lands of the Northwest of Brazil with high precision.

The objective of this work was to design, develop, and evaluate the correct operation of two low-cost pieces of measuring equipment that allowed for obtaining, in real-time, the moisture content of gypsum and plaster materials. For this, on the one hand, monitoring tests were carried out on studying the variation in the mixing water content of the samples, since they were poured in the fresh state in the mold using a resistive sensor in Arduino developed for this purpose and performing complementary thermogravimetric analysis. Moreover, with the help of a capacitive sensor of our design, the evolution in the interior moisture content of the specimens was determined from the demolding to the performance of the standardized tests after seven days, relating these measurements to the evolution of the properties’ mechanics of materials.

## 2. Materials and Methods

### 2.1. Materials

To carry out this work, two differentiated gypsum materials were used for their fineness of grinding: fine white gypsum (YF) for applications in buildings, and plaster (E35) with smaller size particles. The raw material used to obtain the gypsum materials is aljez, a sedimentary rock of chemical precipitation constituted by calcium sulfate with two water molecules (CaSO_4_∙2H_2_O) called gypsum dihydrate (DH). The basis for firing gypsum dihydrate for use as a construction material is carried out with the following basic reaction scheme:(1)$$CaSO_{4}\cdot 2H_{2}O\;\to \;\;CaSO_{4}\cdot \frac{1}{2}H_{2}O\;\left(\alpha ,\beta \right)+\;\frac{3}{2}H_{2}O$$
(2)$$CaSO_{4}\cdot \frac{1}{2}H_{2}O\;\to \;CaSO_{4}+\;\frac{1}{2}H_{2}O$$

In such a way that the dihydrate gypsum is dehydrated according to an endothermic process at temperatures between 105–110 °C to obtain mainly calcium sulfate with 0.5 molecules of water (CaSO_4_∙0.5H_2_O) called gypsum hemihydrate (HH), in its two forms α (more compact and with greater resistance) or β (more soluble and less stable), and which is the majority component of commercial gypsum for building applications. In the implementation of commercial gypsum, mixing with water leads to the rehydration of the HH, giving DH again in the setting process.

For its part, the hemihydrate undergoes endothermic dehydration to obtain anhydrous calcium sulfate (CaSO_4_) called soluble anhydrite or anhydrite III at a temperature below 200 °C [31,32]. The phase transformation of soluble anhydrite to insoluble anhydrite (or anhydrite II), exothermic, and that does not entail associated weight loss, occurs at a temperature, around 200 °C, in the case of hemihydrate and between 300–400 °C for the hemihydrate [32]. The exact temperature at which these dehydration reactions occur depends on experimental conditions, and the separation of events that occur at similar temperatures improves when the rate of heating is low and controlled [33].

The analysis of the raw materials of plaster E35 and fine gypsum YF, supplied by the company Saint Gobain PLACO IBERICA S.A. (Madrid, Spain), was carried out on the powder samples. First, the X-ray diffraction test was carried out using Siemens Krystalloflex D5000 equipment (Madrid, Spain) with a Cu-Kα graphite monochromator. In this way, a diffractogram has been obtained in a range between 5° ≤ 2θ ≤ 100° every 0.04° and one second of passage as can be seen in Figure 1.

In Figure 1, it is observed that both the fine gypsum and the plaster used in this work show four main peaks at angles of 14.7°, 25.7°, 29.7°, and 31.9° corresponding to the hemihydrate gypsum [32]. The presence of anhydrite can be seen in the fine gypsum sample with diffraction peaks located at angles of 25.4° and 31.4°.

Besides, a thermogravimetric analysis of both raw materials was also carried out using TA Instruments SDT Q600 equipment (Madrid, Spain). Heating was carried out from room temperature to 400 °C at a rate of 5 °C/min and air atmosphere (100 mL/min), analyzing an approximate mass of 40 mg in each test. The results obtained for the two gypsum materials used are shown in Figure 2 and Figure 3. They show the evolution of the mass of the samples versus temperature (green line), the first derivative of the mass versus temperature (blue line), and the energy per unit mass versus temperature (brown line) that monitors associated thermal events. The figures show the percentage of mass (%) lost concerning the initial mass and the mass (mg) lost in each event, as well as the temperature at which the maximum speed in the loss of mass occurs.

From the analysis of Figure 2 and Figure 3, it was possible to observe that both the E35 plaster and the YF fine gypsum showed a first loss of mass (corresponding to 0.44% for plaster and 1% for gypsum) due to the surface humidity below 75 °C. Next, the second loss of mass was observed that took place between 75 and 175 °C as a consequence of the dehydration of the hemihydrate that gives rise to soluble anhydrite with a maximum of 131 °C for plaster and 125 °C for gypsum and associated mass losses of 5.8% and 4.2%, respectively; this event is endothermic as can be seen in the curve corresponding to heat flow. The exothermic phase transformation from soluble anhydrite to insoluble anhydrite did not lead to associated weight loss and was observed in plaster at 344 °C and less clearly in gypsum at 357 °C. These temperatures indicate that in both cases the sample contains β-hemihydrate.

The amount of hemihydrate (HH) present in the raw material samples on both a dry and wet basis is presented in Table 1. For the calculation, the initial mass of the sample subjected to thermal analysis, the loss in mass of the second thermal event between approximately 75 and 175 °C, and the stoichiometry of the reaction Equation (2) have been taken into account.

As can be seen in Table 1, the purity of the E35 plaster sample was much higher with a hemihydrate content close to 94%, which was reduced to 67% in the fine gypsum sample (wet base). This was due, on the one hand, to the presence of anhydrite in fine gypsum as shown in the X-ray diffractogram and, on the other hand, to the additive content of fine gypsum raw materials which decomposes at high temperatures (not shown in the thermogravimetric analysis).

On the other hand, Table 2 shows the dosages used for the preparation of the RILEM (The International Union of Laboratories and Experts in Construction Materials, Systems, and Structures) standardized 4 cm × 4 cm × 16 cm test specimens. It should be noted that high water/conglomerate ratios were chosen to show the effect that the variation in the content of mixing water and humidity has on the setting process of the samples.

The preparation of all the batches was carried out with the same technique and equipment following the recommendations of the UNE-EN 13279-1: 2009 [34] standard, always under the same environmental conditions (65% relative humidity and 22 °C ambient temperature) and without performing an energetic kneading that could break the internal microstructure of the gypsum composite materials.

### 2.2. Design and Development of Low-Cost Measurement Equipment

#### 2.2.1. Arduino Resistive Sensor

The low-cost Arduino sensor used in this work consists of the following electronic components: Arduino UNO board with ATmega328P microprocessor with 14 digital inputs/outputs and 6 analog inputs, with the possibility of a USB connection to the computer, 8 GB MicroSD memory module to store the information without saturating the internal memory of the Arduino controller, and a humidity sensor, the Arduino Grove-Moisture Sensor HL-69 model (RS component, Madrid, Spain). Figure 4 shows the sensor connection diagram.

It is worth highlighting the versatility that Arduino has to connect the physical world with the digital world, the possibility of automating the samplings, and the large existing free software community that allows quick access to the information flows about the sensors used [35]. On the other hand, the resistive sensor HL-69 has been used previously in other research works to determine the moisture content of different types of soils [36], basing its operation on the electrical conductivity of the medium in which it is introduced according to the Equation (3):(3)I=σ·E·S
where *I* is the current intensity, *E* is the intensity of the electric field generated by the potential difference between the sensor tips, *S* is the section, and *σ* is the conductivity of the mixture that decreases as the mixing water is lost in the test specimens.

In Figure 5, an explanatory flow diagram of the program developed for monitoring the moisture content inside the gypsum and plaster specimens during the setting process is shown.

As can be seen in Figure 5, the Arduino programming code is divided into three blocks. The first consists of the initialization of variables and constants that will be used in it as well as the recognition of all the devices that intervene in the process and the verification of the sending of information from the sensor to the MicroSD card or the computer’s database. Second, is the automation of the sampling through a previously predetermined data capture period, which for the case of this work was considered five minutes during the seven days established by the UNE-EN 13279-2 standard for gypsum and plaster. The third and last was the creation of a single plain text file per test with the data: number of samples being carried out, percentage of humidity, and time at which the capture was carried out.

Finally, it is necessary to highlight that the total cost of the designed Arduino humidity sensor was approximately 35 euros, this being a competitive price compared to other commercial humidity sensors on the market, such as hygrometers or inductive sensors, which have higher prices [37]. Also, the device used had the advantage of being able to carry out measurements in the variation of the moisture content of the samples from when the gypsum material was poured into the mold once it was kneaded; this presents great industrial advantages since data were obtained from the beginning of the setting of the paste and without the need to unmold the specimens to perform the measurements [28].

#### 2.2.2. Capacitive Sensor

Capacitive sensors that base their operation on the action of an electric field have been used successfully in industry to measure physical parameters. Some authors have used this type of device to measure humidity in construction elements such as wood or cement mortars [23]. This type of sensor has the advantage of being easy to use and economically viable. In this work, a capacitive sensor linked to a self-oscillating circuit, like the one shown in Figure 6, was developed. Using this system, it was possible to measure small variations in the loss of mixing water contained in the gypsum and plaster specimens as they hardened.

In this way, the implemented sensor consists of a flat-parallel plate capacitor with a surface area of $4\times \;16\;{\mathrm{cm}}^{2}$ corresponding to the side of the test specimens and which is part of a self-oscillating RLC (resistance–coil–capacitor) circuit. Copper plates that are conventionally used in the elaboration of printed circuits have been used, with a conductive copper face and the outer face insulated with bakelite to avoid measurement errors that could falsify the experiment. The operation of the measuring equipment was based on finding the resonance frequency of the circuit, that is, the one in which the current intensity that circulates was maximum and the pulsation was zero $\left(X_{L}-X_{c}=0\right)$. This resonant frequency was found by viewing the Lissajous figure corresponding to the composition of two signals that are in phase on the oscilloscope. In this way, according to the determination of said resonance frequency measured with the help of an oscilloscope, it is possible to know the variation in the moisture content of the materials tested employing Equation (4):(4)$$L\omega =\frac{1}{C\omega }\;⟹\;f=\frac{1}{2\pi \sqrt{LC}}\;\;\left[Hz\right]\;\;\;\;\;\;\;with\;\;C=\epsilon \cdot \frac{A}{d}\;\left[F\right]$$
where *f* is the resonance frequency in Hz, *C* is the capacitor capacity in *F*, *L* is the inductance of the sensor in *H*, Є is the permittivity of the dielectric (in this work of the wet specimen), *A* is the surface of the condenser plates m^2^, and *d* is the distance between the plates of 0.04 m corresponding to the thickness of the standard specimens.

However, sometimes it is necessary to take into account factors that can influence the capacitance measurement. This is the case of edge effects that appear when the distance between plates is excessive; for this reason, it is recommended to use the corrected equation to calculate the capacitance:(5)$$C=\varepsilon \cdot \frac{l\cdot a}{d}\cdot \left[1+\frac{l}{\pi }\left[\ln\left(\frac{\pi \cdot a}{d}+1\right)+1+\ln2\right]\right]\;$$
where l  and a are the length and width, respectively, in m of the capacitor plates. Besides, it is advisable to always maintain the same ambient humidity in laboratory conditions to avoid the appearance of parasitic resistances that could harm the measurement. The importance of using shielded cables to avoid capacitive interference in the measuring equipment, and the need to ensure good contact between the copper plate and the gypsum or plaster material, should also be highlighted, thus avoiding the effect of air as a dielectric that could get in the way of taking measurements.

Therefore, it is through the variation in the electrical permittivity of the material caused during the setting that the measurements related to the humidity of the sample are obtained. It is a very useful method that allows knowing the variation in the percentage of humidity in the specimens just before carrying out the mechanical tests, to subsequently establish relationships that allow the two variables to be related. In addition to being an alternative method to the traditional gravimetric tests, the system has a low cost, is easy to use, and its measurement is not only on the surface as are most commercial hygrometers [38].

### 2.3. Methodology

The experimental program carried out in this investigation is presented visually in Figure 7. A total of 24 specimens from the same batch were made for each type of gypsum material and dosage. On the one hand, a test piece of each type was used for monitoring, using the resistive sensor in Arduino, the variation in the mixing water content (100%) until the time of testing according to the UNE-EN 13279-2 standard at 7 days (0%) with a sampling frequency of five minutes. A series of three specimens of each type were tested daily to determine the evolution of the mechanical properties (bending and compression), and the mean value of the three values is presented in this work and their relationship with the moisture content obtained with the help of a capacitive sensor. Finally, complementary tests were also carried out employing gravimetric thermal analysis; these have been carried out on the same specimen at ages of 1, 4, and 7 days in dosages YF-0.8 and E35-0.7 to monitor the composition of the sample and quantify the proportion of dihydrate that is formed at the expense of the hemihydrate as the gypsum sets.

## 3. Results and Discussion

### 3.1. Arduino Sensor Monitoring

Figure 8a shows the measurement equipment implemented in Arduino and responsible for monitoring the variation in the mixing water content of the test specimens, expressed in terms of percentage of humidity. The sampling time was five minutes for seven days, the time specified by the UNE-EN 13279-2 standard as the time necessary to allow the gypsum composite materials to set before testing their mechanical resistance. The sensors were duly calibrated by setting the value of 100% in the measure corresponding to the pouring of the material in the mold and replaced in the measure of the different specimens so as not to distort the results. In addition, and in order to avoid measurement errors, two sensors per specimen were used to counteract the validity of the results, and the mean value of both measurements is reflected in this work.

In parallel, the results obtained for each of the samples tested are shown in Figure 8b. From the analysis of the figure, it can be seen how the variation in the content of mixing water collected by the sensor in the first three days is very scarce, being from that moment on where there is a sharp decrease in the percentage of water in the test specimens. On the seventh day collected by the UNE-EN 13279-2 standard as the age to carry out the mechanical tests, it can be seen how all the samples have reached zero or values close to it; this indicates variations in humidity not measurable by the sensor and the end of hardening of the studied samples. It can also be observed that the higher the mixing water content in the samples, the longer it takes to reach zero, and the specimens made with fine gypsum material took longer to lower their moisture content than the test specimens made with plaster.

In this way, the Arduino system presented was very useful to carry out simple and fast monitoring of the setting process and variation in the mixing water content of the test specimens made with gypsum materials. This monitoring finds an immediate application in industrial companies destined to the elaboration of precast since it allows to check if the hardening process of the samples is developing normally.

In addition to this monitoring, a thermogravimetric analysis was carried out with the help of the same equipment and techniques for powder samples at YF-0.8 and E35-0.7 dosages for test specimens aged 1, 4, and 7 days. As an example, Figure 9, Figure 10, Figure 11 and Figure 12 show the results of the thermogravimetric analysis of samples E35-0.7 at 1 and 7 days of age, and of samples YF-0.8 at 1 and 7 days of age, respectively. In the thermal analysis of all the specimens, three mass losses were observed, all corresponding to endothermic events. The first would correspond to the loss of free water, which occurred at a temperature below 100 °C, and which in the younger specimens, both plaster and gypsum, presents a maximum in the derivative of the mass at approximately 58 °C. This first loss of water is of much lower magnitude in the seven-day-old specimens and did not present a maximum in the curve derived from the mass.

In the temperature interval between 75 and 175 °C, the second mass loss occurs with a maximum in the derivative of the mass around 122–125 °C. This second loss presents a peak with a maximum in the derivative of the mass between 137 and 141 °C which constitutes the third mass loss of the sample. The second loss of mass was due to the dehydration of the dihydrate gypsum formed in the reaction of the hemihydrate with the mixing water; the third loss was due both to the dehydration of the hemihydrate formed in the second mass loss and to the dehydration of the original hemihydrate remaining in the test tube and from the original raw material which gives rise to soluble anhydrite. The exothermic phase change from soluble to insoluble anhydrite can be seen in all the test specimens with a maximum temperature of 351–355 °C.

Table 3 shows the mass loss, as a percentage concerning the original mass, of each of these three events, for the specimens of different ages as well as the total mass loss.

It was observed that the percentage of free water in the plaster specimen decreased from 13.8% in the youngest specimen to a value of 0.3% in the oldest specimen; in the gypsum specimen, these values decreased from 16% to 0.5%. This decrease was due both to the evaporation of the mixing water during drying and to the progressive hydration of the hemihydrate to form dihydrate.

Accordingly, the loss of mass associated with the dehydration of the hemihydrate present in the specimen (second loss in the thermogram) increased in plaster samples from 12.3% to 14.5% with the increasing age of the specimen, while in the gypsum samples, the values evolved from 10.3% to 12.6% at the age of seven days. These values reflect the hydration of the hemihydrate that constitutes the raw materials to produce an increasing amount of dihydrate with the age of the specimen as the setting process occurs.

The amount of dihydrate (DH) and hemihydrate (HH) present in the test specimens were calculated from the second and third mass loss, taking into account the initial mass of each sample, the stoichiometry of reactions (1) and (2), and discounting the mass of hemihydrate formed in reaction (1) in calculating the amount of hemihydrate. The results are presented in the last two columns of Table 3.

In the plaster specimens, the dihydrate represents 78.2% of the youngest specimen and 92.4% in the oldest specimen. These values were, respectively, 65.3% and 80.3% for the fine gypsum specimens. The amount of hemihydrate in the plaster specimens decreased from 9.8% to 5.1% as the set progressed; these values were 0.4% and 0%, respectively, for the fine gypsum specimens.

These results show the evolution in the composition of the specimens throughout the setting process, with the progressive formation of dihydrate, responsible for the mechanical resistance of the gypsum compound as well as the progressive decrease in the free water of the sample (humidity) throughout the process [39]. Also, the results indicate that the plaster specimens present a higher amount of dihydrate than the plaster samples, according to the composition of the raw materials indicated in Table 1, and that it accounts for the higher mechanical strengths for these specimens.

### 3.2. Tests Using the Capacitive Sensor

Figure 13 shows the non-intrusive capacitive sensor developed for taking moisture measurements in the gypsum materials used in this investigation. This sensor allows us to know the variation in the moisture content in the test specimens over time and its determination at the exact moment of the mechanical tests.

To do this, first, and as shown in Figure 14, a sensor calibration has been carried out to determine its behavior and relate the obtained capacitance values with the moisture content of the test piece. Thus, the initial measurement of capacitance collected with the sensor after removing the specimens was taken as a value of 100% in humidity and as a value of 0% the measurement of the specimen dried in an oven at a temperature of 50 ± 2 °C for 48 h. Said capacitances have been related to the determination of the humidity obtained by gravimetric methods according to the following Equation (5):(6)$$M=\frac{W_{nd}-W_{0}}{W_{0}}\cdot 100\;\left[\%\right]$$
where $W_{nd}$ is the humidity on day *n* studied and W_0 is the weight of the test piece in an anhydrous state dried in an oven.

As can be seen in Figure 14, the capacitance values oscillate around 150 and 350 pF for the gypsum materials tested. You can see how there is a strong positive correlation when fitting the values using second-degree polynomial equations. At high humidity values, there are differences between the capacitance values among all the tested samples, observing higher capacitance values for the samples with the highest water/conglomerate ratio of each typology. For humidity values lower than approximately 50%, a single curve was obtained regardless of the material treated. Furthermore, it can be seen that the higher the water/conglomerate ratio, the higher the capacitance values are obtained regardless of the material treated.

Figure 15 and Figure 16 show the results derived from the flexural and compression rupture tests at different ages concerning the moisture content of the specimen, carried out daily from day one after removal from the mold until reaching seven days. The techniques and recommendations of the UNE-EN 13279-2 [40] standard have been followed and a hydraulic press of the Autotest-200/10-SW model have been used to carry out the mechanical tests. Furthermore, all laboratory tests have been carried out under the same conditions of relative humidity and temperature (50 ± 5% and 22 ± 2 °C).

As can be seen after the analysis of Figure 15 and Figure 16, the evolution of the mechanical strengths of the gypsum material test specimens tested develops, as there is a variation in the moisture content measured with the help of the capacitive sensor developed. It can be seen how the maximum levels of resistance are reached at the age of seven days set by the reference regulations and corresponding to the minimum levels of humidity determined with the equipment implemented. In turn, there is a significant change in the evolution of resistance in the transition between the fourth and fifth days, this period being in which the variations in water content of the test specimens have been more pronounced. Finally, and following other previous studies [41], the plaster material specimens have shown better performance in terms of mechanical resistance than their homologs made with gypsum, presenting higher resistance values than those that contained less amount of mixing water in origin.

In addition to the tests carried out, the analysis by X-ray diffraction was carried out on the test pieces YF-0.8 and E35-0.7. The results obtained are shown in Figure 17.

X-ray diffraction analysis for the specimens hardened at the age of seven days (Figure 17) shows diffraction peaks located at angles of 11.6°, 20.7°, 29.1°, 23.4°, and 31.1° corresponding to dihydrate (DH), and did not show peaks corresponding to hemihydrate (HH). The diffractograms are practically coincident between the YF-0.8 and E35-0.7 specimens with the only exception that the diffraction peaks were more intense for the specimen made with plaster material.

Finally, in Table 4, the confidence intervals for humidity, capacitance, flexural strength, and compressive strength were calculated for each day of measurements. These intervals were calculated to contain at least 75% of the data because the hypothesis of normality cannot be verified.

## 4. Conclusions

In this work, two low-cost sensors were developed capable of determining in real-time the variation in mixing water content experienced by gypsum materials during their hardening. In this way, it was possible to delve into the study of the setting of this type of construction material and its relationship with the evolution of their mechanical resistance to bending and compression.

On the one hand, an Arduino resistive sensor was used that made it possible to monitor in real-time the percentage variation in the moisture content of the test tube, since the fresh mix material was poured into the mold. This inexpensive system is easily usable and can be used to determine the state of the materials located in situ at the site, as well as to study the evolution of the humidity that may appear in rehabilitation projects. The effect of increasing the water/conglomerate ratio during the hardening process was also verified, since it not only translates into an increase in the workability of the mixture but also produces slower hardening of the samples. Complementary to this study, XRD analyzes and thermogravimetric analyses were performed at different ages to visualize the evolution of the hemihydrate and dihydrate compounds typical of gypsum materials.

On the other hand, a capacitive sensor based on obtaining the resonance frequency of a self-oscillating RLC circuit was implemented. The results showed how the designed sensor allows for measuring the moisture content of the gypsum and plaster samples in a reliable and non-intrusive way. These characteristics have made it possible to relate the variation in mixing water content of the test specimens with the evolution of mechanical resistance. As has been proven, both in the case of flexural and compressive strength, these increase as the moisture content of the test tube decreases until reaching the maximum value at the age of seven days, a date set by the UNE-EN standard, of which UNE-EN standard 13279-2 is the optimal one for conducting mechanical tests. It was verified that the plaster samples showed higher resistance values than the gypsum specimens. This can be attributed, in part, to the higher proportion of gypsum hemihydrate present in E35 compared to YF, which was revealed by thermogravimetric analysis. Thus, this sensor has a strong application for the determination of the evolution in the moisture content and its relationship with the mechanical resistance in different construction elements, as long as the positioning of the condenser plates and the separation of these is not possible or greater than 20 cm (e.g., interior partitions of dry partition or plates for false ceilings).

As future lines of research, it would be good to test the operation of this type of sensor for measuring humidity in situ. Moreover, it would be convenient to carry out tests with other types of construction materials such as lime, mortar or concrete. On the other hand, in materials, such as gypsum and plaster, an exothermic reaction occurs during the setting. Also, it is intended to monitor this internal temperature of the samples and relate it to the variations in the content of mixing water in the specimens to improve understanding of the phenomenon studied.

## Figures and Tables

**Figure 1 materials-13-05831-f001:**
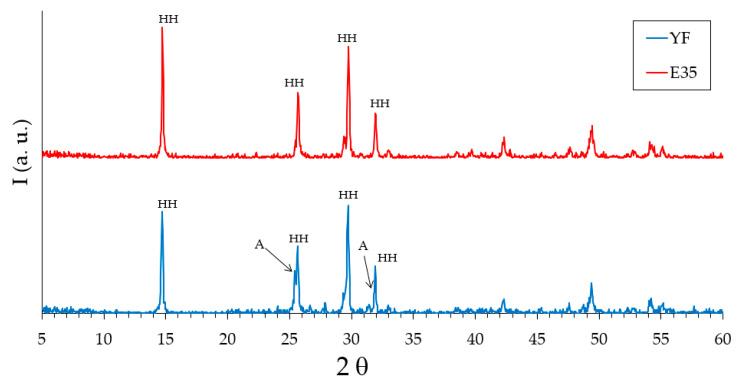
Analysis by X-ray diffraction (XRD) of the raw materials. HH: hemihydrate. A: anhydrite.

**Figure 2 materials-13-05831-f002:**
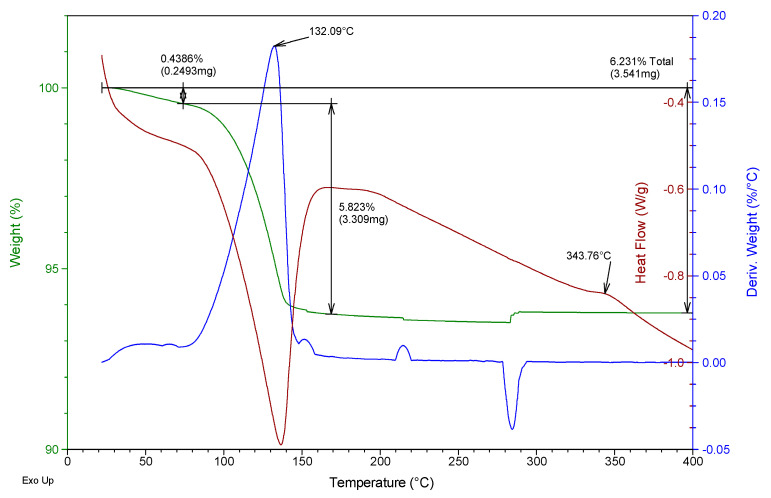
Thermogravimetric analysis of the E35 plaster used.

**Figure 3 materials-13-05831-f003:**
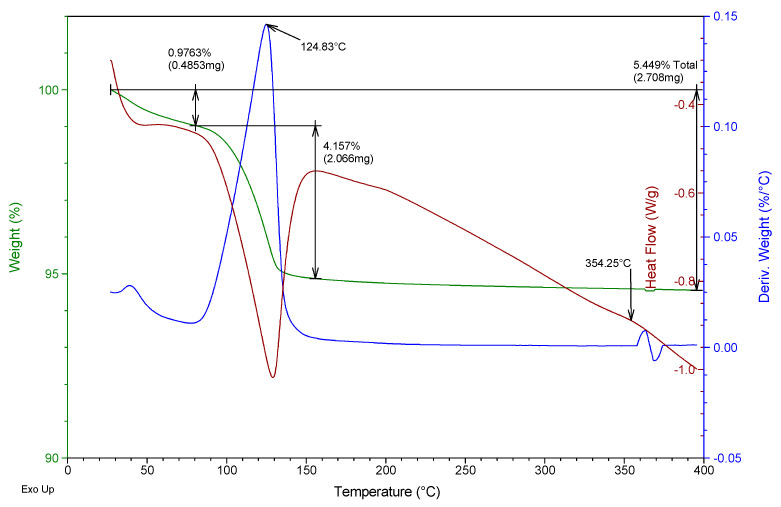
Thermogravimetric analysis of the fine gypsum YF used.

**Figure 4 materials-13-05831-f004:**
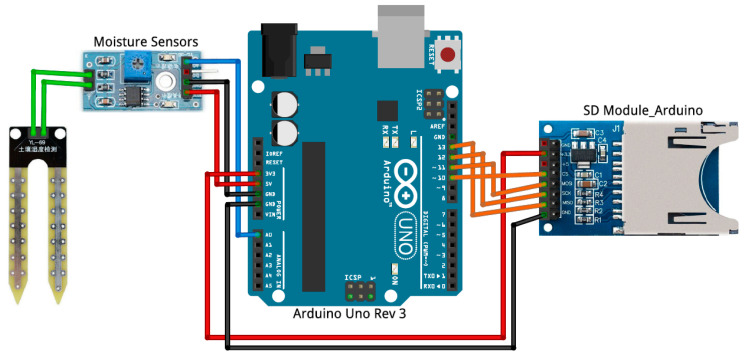
Arduino sensor connection diagram.

**Figure 5 materials-13-05831-f005:**
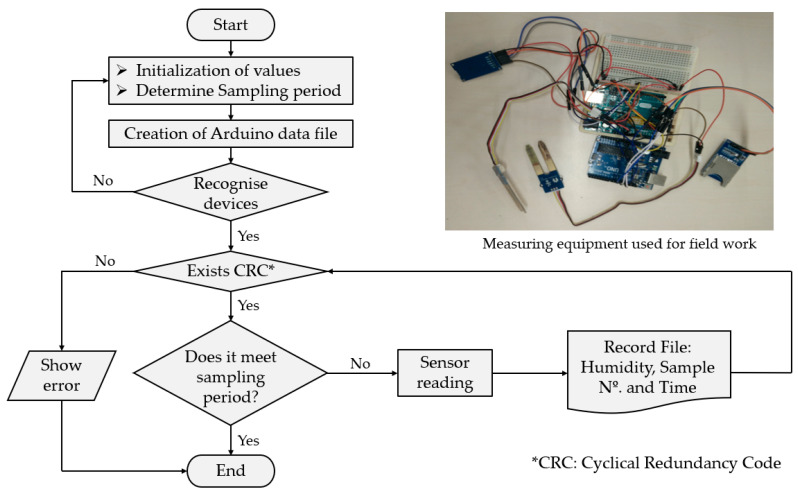
Explanatory flow diagram of the program developed for data capture.

**Figure 6 materials-13-05831-f006:**
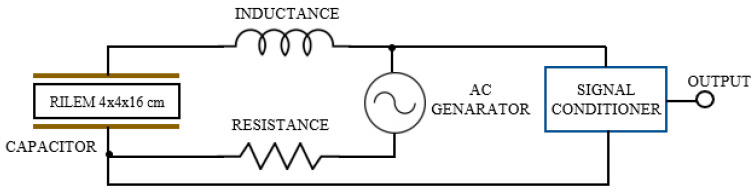
Diagram of the capacitive sensor developed for taking measurements. RILEM: (The International Union of Laboratories and Experts in Construction Materials, Systems, and Structures)

**Figure 7 materials-13-05831-f007:**
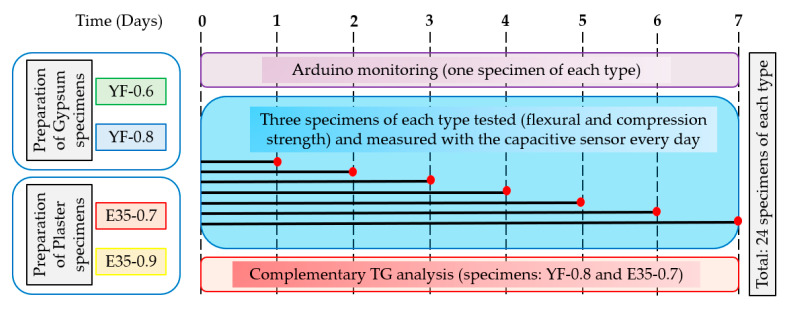
Experimental program and tests carried out.

**Figure 8 materials-13-05831-f008:**
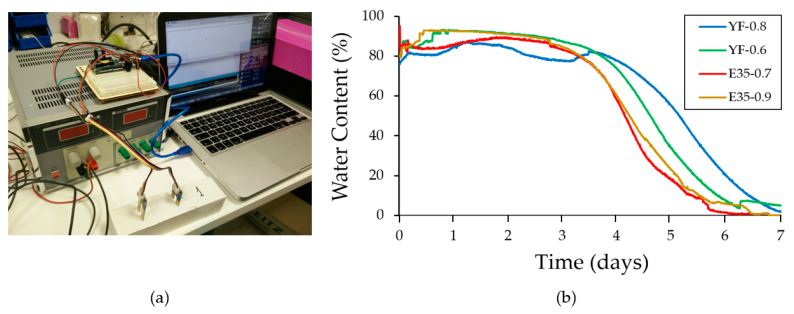
Testing using the Arduino sensor. (**a**) Measurement equipment monitoring a plaster specimen; (**b**) results obtained from the variation in the mixing water content against time for each of the tested specimens.

**Figure 9 materials-13-05831-f009:**
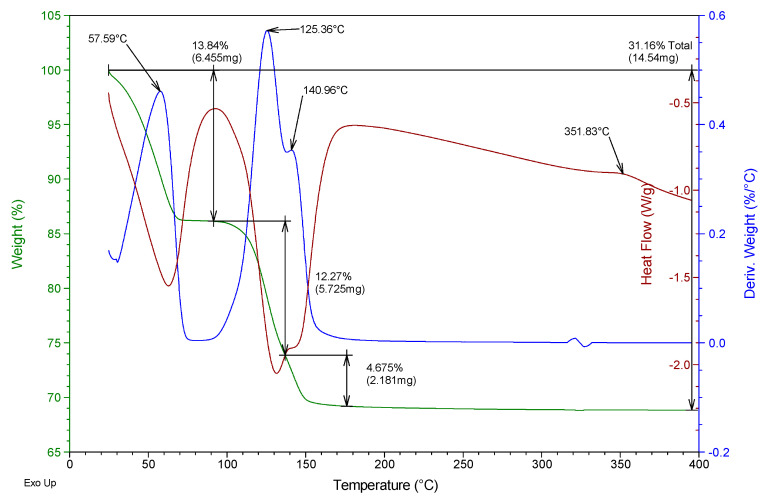
Thermogravimetric analysis of the test tube E35-0.7 aged 1 day.

**Figure 10 materials-13-05831-f010:**
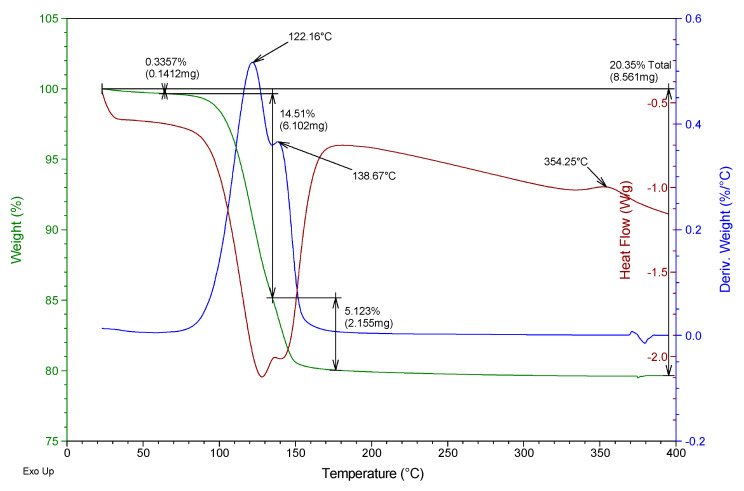
Thermogravimetric analysis of the E35-0.7 test tube aged 7 days.

**Figure 11 materials-13-05831-f011:**
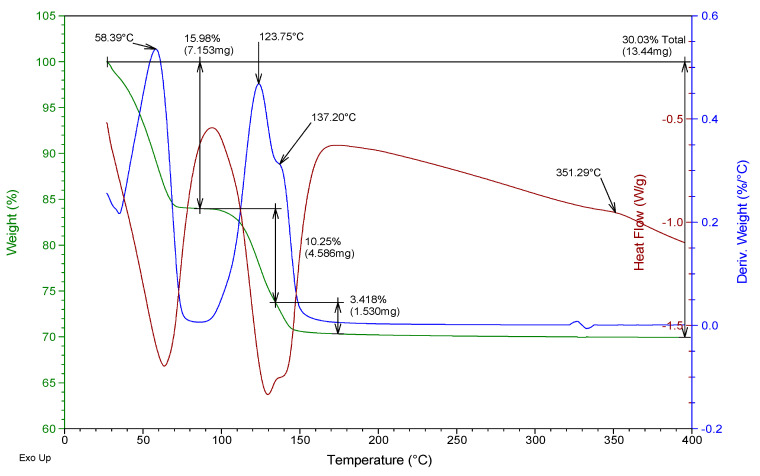
Thermogravimetric analysis of the test tube YF-0.8 aged 1 day.

**Figure 12 materials-13-05831-f012:**
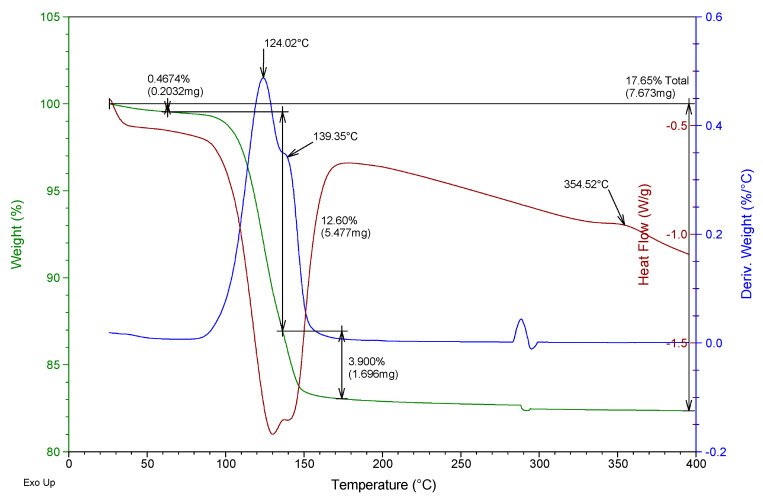
Thermogravimetric analysis of the YF-0.8 test tube aged 7 days.

**Figure 13 materials-13-05831-f013:**
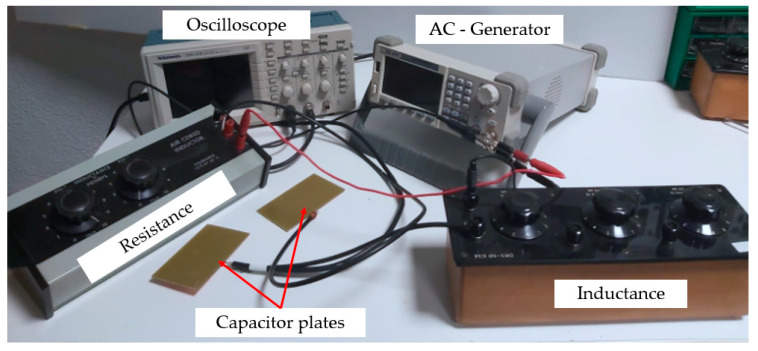
Capacitive measurement equipment used.

**Figure 14 materials-13-05831-f014:**
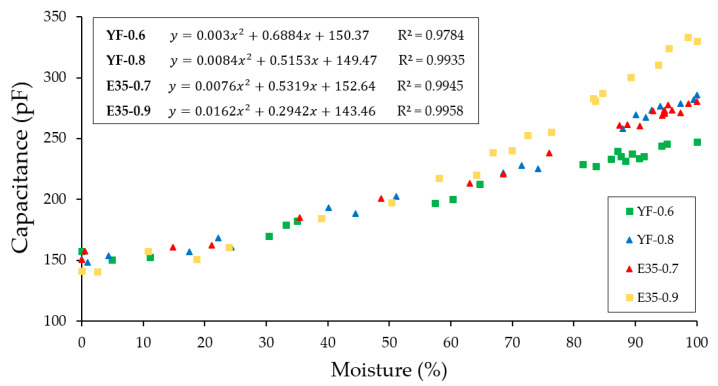
Capacitance values versus the moisture content of the test specimens.

**Figure 15 materials-13-05831-f015:**
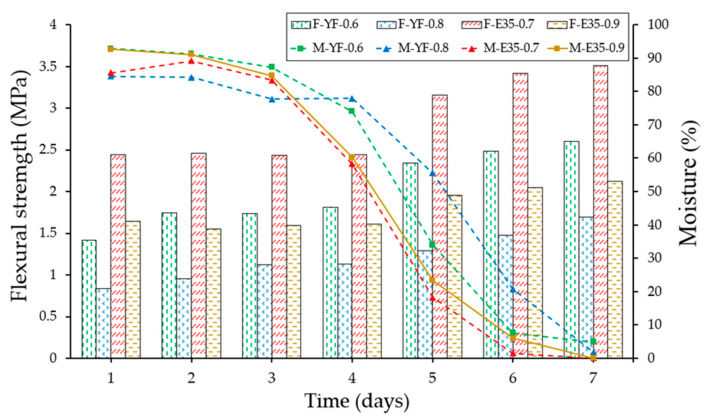
Evolution of flexural strength (F) and moisture content (M) of the different specimens over time.

**Figure 16 materials-13-05831-f016:**
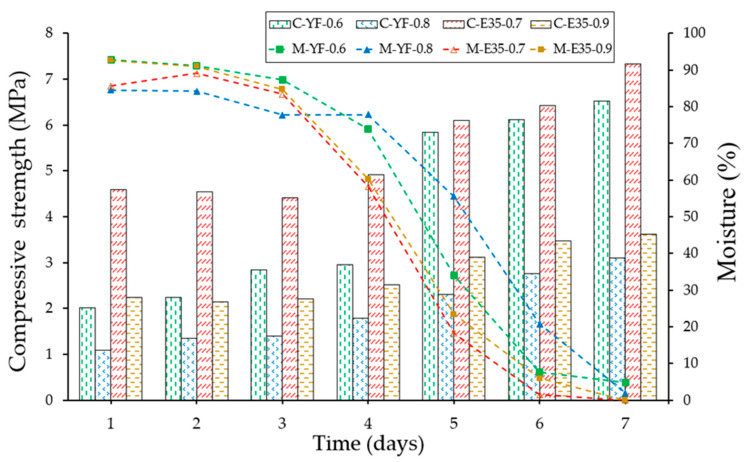
Evolution of the compressive strength (bars) and the moisture content (lines) of the different specimens against time.

**Figure 17 materials-13-05831-f017:**
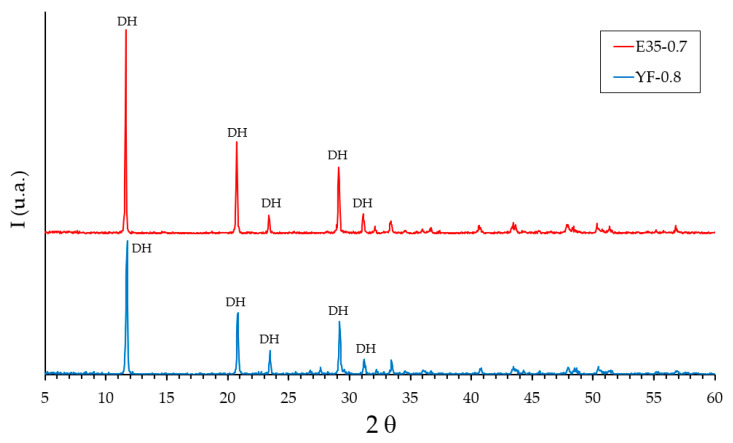
Results of the analysis by X-ray diffraction of the test specimens YF-0.8 and E35-0.7 at the age of 7 days. DH: dihydrate.

**Table 1 materials-13-05831-t001:** Results of the thermal analysis of raw materials.

Sample	$\%\;H_{2}O$	% HH Wet Base	% HH Dry Base
E35	0.4	93.85	94.27
YF	1.0	67.00	67.66

**Table 2 materials-13-05831-t002:** Dosages used for mixing the test specimens.

Denomination	Water (g)	Gypsum (g)	Denomination	Water (g)	Plaster (g)
YF-0.6	600	1000	E35-0.7	700	1000
YF-0.8	800	1000	E35-0.9	900	1000

**Table 3 materials-13-05831-t003:** Results of the thermogravimetric analysis for samples YF-0.8 and E35-0.7.

Sample *	Mass Loss				Composition	
	*Total* Δ*m (%)*	Δ*m H_2_0 (%)*	Δ*m DH (%)*	Δ*m HH (%)*	*%DH*	*%HH*
E35-0.7-1	31.2	13.8	12.3	4.7	78.2	9.8
E35-0.7-4	20.4	0.3	14.4	5.2	91.9	6.7
E35-0.7-7	20.4	0.3	14.5	5.1	92.4	5.1
YF-0.8-1	30.0	16.0	10.3	3.4	65.3	0.4
YF-0.8-4	18.1	0.5	12.3	4.2	78.6	2.3
YF-0.8-7	17.7	0.5	12.6	3.9	80.3	0.0

* The numbers 1, 4, and 7 indicate the age in days that the sample was tested.

**Table 4 materials-13-05831-t004:** Confidence intervals for moisture, capacitance, flexural strength, and compressive strength measured daily on each sample.

**Intervals for Humidity Expressed as a Percentage**				
**Day**	**YF-0.6**	**YF-0.8**	**E35-0.7**	**E35-0.9**
1	(90.10, 100)	(96.05, 100)	(96.07, 100)	(93.27, 100)
2	(88.56, 92.49)	(91.58, 96.04)	(94.1, 96.5)	(80.1, 98.3)
3	(86.57, 88.97)	(86.21, 93.65)	(92.02, 95.98)	(72.86, 89.14)
4	(79.00, 88.40)	(65.83, 77.03)	(85.53, 92.41)	(64.08, 75.32)
5	(53.44, 68.16)	(34.15, 56.31)	(56.24, 82.16)	(43.79, 71.27)
6	(28.23, 37.51)	(14.36, 28.24)	(7.67, 62.47)	(6.19, 48.15)
7	(0, 16.56)	(0, 6.42)	(0, 21.9)	(0, 15.75)
Ranges for capacitance expressed in pF				
Day	YF-0.6	YF-0.8	E35-0.7	E35-0.9
1	(242.29, 248.73)	(274.69, 289.65)	(266.52, 286.80)	(320.25, 338.33)
2	(231.96, 239.04)	(269.63, 278.75)	(269.47, 280.11)	(276.26, 322.5)
3	(227.68, 234.44)	(252.94, 276.82)	(267.02, 274.70)	(242.49, 303.69)
4	(223.35, 236.15)	(219.29, 230.89)	(260.09, 261.85)	(228.20, 259.56)
5	(186.64, 219.92)	(180.48, 208.68)	(198.34, 249.46)	(187.22, 236.38)
6	(164.48, 189.72)	(150.41, 173.54)	(144.03, 221.31)	(130.75, 199.75)
7	(145.97, 161.01)	(145.57, 156.21)	(145.64, 166.64)	(127.74, 165.10)
Intervals for flexural strength expressed in MPa				
Day	YF-0.6	YF-0.8	E35-0.7	E35-0.9
1	(1.26, 1.58)	(0.75, 0.95)	(2.32, 2.55)	(1.56, 1.72)
2	(1.67, 1.83)	(0.91, 0.99)	(2.42, 2.50)	(1.49, 1.61)
3	(1.68, 1.80)	(0.96, 1.28)	(2.35, 2.51)	(1.41, 1.77)
4	(1.70,1.94)	(1.03, 1.23)	(2.32, 2.56)	(1.55, 1.67)
5	(2.20, 2.48)	(1.21, 1.37)	(3.08, 3.24)	(1.85, 2.05)
6	(2.38, 2.58)	(1.40, 1.56)	(3.26, 3.58)	(1.99, 2.11)
7	(2.54, 2.66)	(1.61, 1.73)	(3.43.3.59)	(2.04, 2.20)
Intervals for compressive strength expressed in MPa				
Day	YF-0.6	YF-0.8	E35-0.7	E35-0.9
1	(1.94, 2.10)	(0.91, 1.27)	(4.41, 4.77)	(2.12, 2.35)
2	(2.04, 2.44)	(1.05, 1.65)	(4.30, 4.78)	(1.95, 2.35)
3	(2.56, 3.12)	(1.21, 1.57)	(4.63, 4.49)	(2.09, 2.41)
4	(2.80, 3.14)	(1.58, 1.98)	(4.69, 5.13)	(2.25, 2.77)
5	(5.69, 6.01)	(2.17, 2.45)	(5.94, 6.28)	(3.02,3.22)
6	(5.92, 6.32)	(2.64, 2.88)	(6.28, 6.56)	(3.21, 3.70)
7	(6.36, 6.36)	(3.00, 3.20)	(7.14,7.54)	(3.49, 3.77)

Note that at least 75% of the data are within the calculated range.

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
