# Peer review of "Low-Cost Sensors for Determining the Variation in Interior Moisture Content in Gypsum Composite Materials"

_materials, 2020, doi:10.3390/ma13245831_

Round 1
Reviewer 1 Report
Dear editor,
This study is about how to measure the amount of humidity contained in Gypsum Composite Materials using two newly developed low-cost sensors. Overall, the test method and procedure were clearly established, and the contents were judged to be novel. Therefore, if the following questions are answered and minor revisions are made, this paper is considered to be ready for publication.
- Please correct the error part of the overall paper.
For example:
Line 109 : ?-hemihydrate
Line 126 : Figures 2 y 3 (y means?)
Line 193, 268 : database. . (double period) etc.
- Please write the full name for the first abbreviation that appears throughout the paper.
For example:
Line 156 : RILEM
Line 163 : UNE-EN
Line 217 : RLC etc.
- How much is the difference between the water content measured in Figure 8 and the actual water content? I am curious about the reliability of Arduino Sensor Monitoring.
- The sensor used in this paper is expressed as a low cost, but it would be better to mention in the manuscript how much cheaper compared to a sensor used in general.
Author Response
The authors would like to thank the reviewer for their comments and praise for the work done. All questions have been answered and the document has been corrected based on your successful revisions.
- Please correct the error part of the overall paper.
For example:
Line 109: hemihydrate
Line 126: Figures 2 and 3 (and means?)
Line 193, 268: database. . (double period) etc.
The errors noted and others found in the document have been corrected.
- Please write the full name for the first abbreviation that appears throughout the paper.
For example:
Line 156: RILEM
Line 163: UNE-EN
Line 217: RLC etc.
Some acronyms have been included, although some, which belong to building regulations, are usually included abbreviated in all scientific articles.
- How much is the difference between the water content measured in Figure 8 and the actual water content? I am curious about the reliability of Arduino Sensor Monitoring.
It is a resistive sensor, so its response depends on the electrical conductivity of the material, which increases as the water content is higher. However, not all this water is free in the material and is used in electrical conduction, much of it combines with the plaster during setting and another part evaporates. For this reason, it would not be entirely accurate to say that the sensor measures the actual water content of the specimen since to know what part of that water has been used to harden the conglomerate material it is necessary to carry out chemical tests such as TGA (Thermal Gravimetric Analysis).
It is a very interesting question and we hope we have clarified your doubt.
- The sensor used in this paper is expressed as a low cost, but it would be better to mention in the manuscript how much cheaper compared to a sensor used in general.
It has been specified in the document that the cost of the sensor is € 35. It can be easily verified that a commercial hygrometer with optimal sensitivity would exceed € 300, so we are talking about an 80% price reduction.
Reviewer 2 Report
Dear Editor,
the topic of the article does not belong to the Materials journal. It is more suitable for the Sensors journal.
Author Response
The authors would like to acknowledge the reviewer's comment
Reviewer 3 Report
The authors present an interesting work on low-cost sensors for determining the variation in interior moisture content in gypsum composite materials. In the reviewer's opinion, this is a relevant work, which provides interesting findings, that deserves to be shared with the scientific community. However, and in contrast to the extensive scientific efforts, the presentation of the manuscript does not comply with the standards of a publication such as Materials MDPI. Therefore, the reviewer suggests the authors to prepare a corrected version by carrying out an extensive edition based on the recommendations provided below. The following suggestions and comments should be taken into account before accepting the article for publication:
- Please show the scatters of the experimental results (I expect that more than one specimen per tests - compression, electrical, etc. - have been considered: I suppose at least three). This applies to Figs. 14, 15 and 16 via error bars and also in the related Tables (by adding the +/- variabilities).
Figure 14. Capacitance values versus moisture content of the test tubes.
Results should include the measurement uncertainty, and expressed as mean value +/- (1x or 2x) standard deviation.
Figure 15. Evolution of flexural strength (F) and moisture content (M)
Figure 16. Evolution of the compressive strength (bars) and the moisture content (lines)
Please discuss the differences between the results considering the statistical significance (e.g. using t-test).
2. What kind of parallel plates have you used for capacitor measurements?
Please state all the details about the capacity measurement setup.
3. You worked in non-intrusive mode, right?
How do you assure a good contact between the metal plate electrode and the mortar sample?
4. How do you deal with the temperature increase due to the current flow in electrical (resistive and capacity) measurements?
5. Test tubes -> prisms. Which tubes?; this should be 4x4x16 cm prismatic specimens, right.
6. In Figure 17. (Results of the Analysis by X-ray Diffraction of the test tubes YF-0.8 and E35-0.7)
please use the square of Intensity in order to better see the smaller peaks.
7. Please also add future research steps which will follow this work.
Author Response
The authors would like to acknowledge the reviewer's comments that have helped significantly improve the scientific quality of the article, including several very interesting contributions to the publication.
The authors present an interesting work on low-cost sensors for determining the variation in interior moisture content in gypsum composite materials. In the reviewer's opinion, this is a relevant work, which provides interesting findings, that deserves to be shared with the scientific community. However, and in contrast to the extensive scientific efforts, the presentation of the manuscript does not comply with the standards of a publication such as Materials MDPI. Therefore, the reviewer suggests the authors to prepare a corrected version by carrying out an extensive edition based on the recommendations provided below. The following suggestions and comments should be taken into account before accepting the article for publication:
- Please show the scatters of the experimental results (I expect that more than one specimen per tests - compression, electrical, etc. - have been considered: I suppose at least three). This applies to Figs. 14, 15 and 16 via error bars and also in the related Tables (by adding the +/- variabilities).
Indeed, at least three specimens have been used as indicated in the article.
Figure 14. Capacitance values versus moisture content of the test tubes.
Figure 15. Evolution of flexural strength (F) and moisture content (M)
Figure 16. Evolution of the compressive strength (bars) and the moisture content (lines)
Results should include the measurement uncertainty, and expressed as mean value +/- (1x or 2x) standard deviation.
Confidence intervals for humidity, capacitance, flexural strength, and compressive strength have been calculated for each day. These intervals are calculated to contain at least 75% of the data because the hypothesis of normality cannot be verified.
Please discuss the differences between the results considering the statistical significance (e.g. using t-test).
The mean values ​​of each measure have not been compared due to not being able to verify the normality of the data.
- What kind of parallel plates have you used for capacitor measurements?
Please state all the details about the capacity measurement setup.
Reviewer comments have been added to the document.
For the measurement, copper plates of those conventionally used in the elaboration of printed circuits have been used, with a conductive copper face and the outer face insulated with bakelite to avoid errors in the measurement that could falsify the experiment. It has been added to the document.
As indicated in the article, the capacity measurement consists of determining the resonance frequency of the circuit. For this, a signal generator is used, and with the help of an oscilloscope, the Lissajous figure corresponding to a resonant circuit is searched, to later be able to relate the capacitance measurement with the variation in the content of mixing water.
- You worked in non-intrusive mode, right?
How do you assure a good contact between the metal plate electrode and the mortar sample?
We understand that the reviewer is referring to the capacitive method. In general, the self-weight of the plate is sufficient for them to make contact and for it to be readable by the oscilloscope. However, sometimes when the measurement is not as refined as desired, a small weight (non-conductor material) can be used on the upper plate to maintain contact (since the lower plate is always in contact due to the specimen's weight).
- How do you deal with the temperature increase due to the current flow in electrical (resistive and capacity) measurements?
The electric current does not influence the temperature of the test piece. In the case of the capacitive sensor, only the variations in the resonant frequency in the RLC circuit are measured and therefore no temperature increase occurs. If the resistive sensor of Arduino is considered, this is a passive component that does not cause temperature increases that influence the material, since it works at a voltage of 3.3 V and the current is only 35 mA and is not high enough to produce increases of temperature.
- Test tubes -> prisms. Which tubes?; this should be 4x4x16 cm prismatic specimens, right.
The expression has been corrected as indicated by the reviewer.
- In Figure 17. (Results of the Analysis by X-ray Diffraction of the test tubes YF-0.8 and E35-0.7)
please use the square of Intensity in order to better see the smaller peaks.
If the values ​​are squared, the entire graph is scaled in the same way, and therefore the smallest and least significant points remain unappreciated. However, for the treated material in question, it is the first five peaks that are conventionally used to identify it.
- Please also add future research steps which will follow this work.
Future lines of research have been added to the end of the document as indicated by the reviewer.
Round 2
Reviewer 3 Report
The methodology of the capacitance measurements needs further elaborations.
Please discuss the cupper-material interface effects on the measured capacity. You are saying that you measure the capacitance of the material, but I would expect that in fact you are measuring the capacity of the copper-material interface. Copper reacts with the material and this could also falsify the measurements. Please discuss all these effects in detail.
Author Response
The methodology of the capacitance measurements needs further elaborations.
Please discuss the cupper-material interface effects on the measured capacity. You are saying that you measure the capacitance of the material, but I would expect that in fact you are measuring the capacity of the copper-material interface. Copper reacts with the material and this could also falsify the measurements. Please discuss all these effects in detail.
The explanations required by the reviewer have been made.